# Discovery–Versus Hypothesis–Driven Detection of Protein–Protein Interactions and Complexes

**DOI:** 10.3390/ijms22094450

**Published:** 2021-04-24

**Authors:** Isabell Bludau

**Affiliations:** Department of Proteomics and Signal Transduction, Max Planck Institute of Biochemistry, 82152 Martinsried, Germany; bludau@biochem.mpg.de

**Keywords:** protein complexes, protein-protein interactions, interactomics, mass-spectrometry, targeted proteomics, data analysis, databases, systems biology

## Abstract

Protein complexes are the main functional modules in the cell that coordinate and perform the vast majority of molecular functions. The main approaches to identify and quantify the interactome to date are based on mass spectrometry (MS). Here I summarize the benefits and limitations of different MS-based interactome screens, with a focus on untargeted interactome acquisition, such as co-fractionation MS. Specific emphasis is given to the discussion of discovery- versus hypothesis-driven data analysis concepts and their applicability to large, proteome-wide interactome screens. Hypothesis-driven analysis approaches, i.e., complex- or network-centric, are highlighted as promising strategies for comparative studies. While these approaches require prior information from public databases, also reviewed herein, the available wealth of interactomic data continuously increases, thereby providing more exhaustive information for future studies. Finally, guidance on the selection of interactome acquisition and analysis methods is provided to aid the reader in the design of protein-protein interaction studies.

## 1. Introduction

Proteins are the main functional molecules within the cell. However, most proteins do not act in solitude. Instead, they often interact with other proteins to form large macromolecular assemblies, which coordinate and perform the majority of molecular functions inside the cell. Many protein complexes are evolutionarily conserved [1] and their dysregulation is associated with the development and progression of various diseases [2]. Identifying and quantifying protein-protein interaction (PPI) networks and stable protein complexes is thus a major goal in basic and translational research.

Importantly, this review mainly focuses on the physical interactome. Thus, a PPI generally refers to a physical interaction between two proteins, unless otherwise stated. A PPI network consequently represents a set of proteins that are connected to each other via PPIs. A protein complex, in turn, refers to a stable group of physically interacting proteins. However, the physicochemical boundaries for differentiating a stable protein complex from more dynamic functional modules in the cell still remain to be defined. In context of this review, clusters of co-regulated PPIs are termed protein complexes.

Over the last decades, mass spectrometry (MS)-based proteomics has emerged as the main approach for studying both the proteome [3,4] and its interactome [5]. Here, I summarize the main MS-based approaches for acquiring data that provides information about PPIs and protein complexes, with an emphasis on the scope of protein complexes covered by each technique. I subsequently focus on computational concepts for analyzing global interactome studies, ranging from discovery- to hypothesis-driven concepts. Finally, I review state-of-the-art databases that can be leveraged when performing complex- or network-centric analyses and provide guidance on the selection of appropriate interactome acquisition and analysis approaches depending on the research question. 

## 2. Targeted and Untargeted Interactome Screens

MS-based workflows for acquiring interactome data can broadly be categorized into two groups: (1) targeted approaches, which focus on a single protein and its interactions at a time, and (2) untargeted interactome screens, which aim to investigate the entire compendium of protein complexes in a given sample. Below I summarize different strategies of each category, focusing on the benefits and limitations of each approach. Figure 1 further provides an overview of the discussed methods, setting each approach in context to its theoretically achievable scope.

The most common MS based technology to study PPIs is affinity purification coupled to MS (AP-MS) [6,7]. Here, a single protein of interest (bait) is purified from a cell lysate by using an antibody that is either specific for the bait or, more commonly, specific for an affinity tag that is fused to the bait. Mild, non-denaturing conditions during the pull-down ensure that stable PPIs between the bait and its interaction partners (preys) are preserved. Bait and prey proteins are subsequently identified and quantified by bottom-up mass spectrometry. This generates a minimal PPI network of a single central node (bait) with all co-purified interaction partners (preys) connected via edges. Importantly, a single AP-MS experiment can potentially determine all stable interaction partners of a given bait. However, it is not possible to directly determine from a single experiment whether all these interaction partners occur in the same macromolecular complex, or if multiple alternative modules with different subunits exist. Addressing this question requires multiple reciprocal pull-downs. Applications of AP-MS range from targeted analysis of a specific macromolecular machinery, to analyses of entire pathways or classes of proteins (e.g., kinases [8]), and to even larger scale studies, such as the effort to map PPIs across all human ORFs [6,9,10]. While the sensitivity and specificity of AP-MS is considered superior to most other techniques and established protocols and data analysis workflows are available, it has three main limitations: (1) a dependency on the availability of specific antibodies, or a requirement for genetic engineering, (2) the necessity to carefully select appropriate controls to ensure that high confidence prey proteins can be distinguished from potentially high background noise [7], and (3) a very high cost for global, and especially differential, analyses. Additionally, only stable interactions that remain intact during sample processing are detectable by AP-MS.

To capture more transient interactions, proximity labelling approaches such as BioID [11] and APEX [12] have recently gained increasing attention. The key concept is that a bait protein is fused to an enzyme, for example a biotin ligase in BioID or a peroxidase in APEX. Upon activation, the enzymes can label proximal proteins at high spatial (~10 nm) and temporal resolution (~1 min labelling speed for APEX) [13,14]. Similar to AP-MS, proximity labelling is a targeted interactome acquisition strategy where experiments are performed on a per-bait basis, meaning that a large number of experiments are required to generate a holistic view of all proximal interactions in the cell, or even across conditions. As the name suggests, the main difference between proximity labelling and other techniques is that the reported prey proteins do not necessarily interact physically with the bait, but they may simply be in close proximity. A recent study applied BioID to generate a proximity map of the human HEK293 cell line, including 35,902 unique high confidence proximity interactions among 4145 spatially localized proteins [15]. Similar to AP-MS, appropriate controls are critical to ensure that high confidence prey proteins can be distinguished from potentially high background labelling.

As discussed above, AP-MS and proximity labelling are inherently targeted approaches, which can only create whole interactome information through combination of a large number of experiments. In contrast, there are a number of methods that are directly geared towards more holistic, untargeted measurements of the interactome, including cross-linking or co-fractionation coupled to MS, thermal proteome profiling and full proteome co-variation analysis. For these techniques, the coverage (number of protein complexes identified) depends solely on the optional application of prior purification steps and technical limitations of the MS instrument and acquisition itself.

In cross-linking coupled to MS (XL-MS), a cross-linking reagent is added to the sample, which covalently links amino acid residues (commonly lysines) that are close to each other. Traditionally, XL-MS has mainly been applied to purified proteins and protein complexes, with the goal to gain structural information [16]. However, recent studies have demonstrated that XL-MS can also be applied on a more global scale, for example investigating interactions between all proteins in the human HeLa cell line [17] or even the entire *Drosophila melanogaster* embryo [18]. Currently, the main limitation of system-wide XL-MS studies is the modest depth of interactome coverage, reaching only a maximum of ~10,000 cross-links among a few hundred proteins, which adds up to ~1000 PPIs [19]. Additionally, the reported interactions are strongly biased towards highly abundant proteins. These effects occur due to technical limitations in MS acquisition and the chemical properties of currently available cross-linkers [19,20].

An alternative approach to gain holistic interactome information in an untargeted fashion is based on complex co-fractionation coupled to MS (CoFrac-MS). Originally developed and applied with the goal to map proteins to different cellular organelles [21,22], CoFrac-MS is now an increasingly popular strategy to probe PPIs and protein complexes on a proteome-wide scale, by utilizing high-resolution fractionation strategies [23,24,25]. Here, protein complexes are extracted under mild conditions to keep them intact, followed by separation and fractionation according to some physicochemical property. Exemplary co-fractionation techniques include size-exclusion chromatography (SEC), which separates complexes by hydrodynamic radius, and ion-exchange chromatography (IEX), which separates complexes by charge. While CoFrac-MS can be applied to purified samples of a sub-proteome (for example, a specific organelle), it is more frequently applied to whole cell lysates. In this case, interactome coverage depends on the resolution of the chromatographic separation, as well as the proteome coverage achieved by the selected MS acquisition of each sampled fraction. Finally, the sensitivity and selectivity of protein complex information obtained by CoFrac-MS are determined by the applied computational inference strategy, which is used to differentiate random co-elution from signals indicating true physical interactions (see next section). In recent years, CoFrac-MS has been widely applied in different contexts, including human interactome maps generated from multi-dimensional fractionation [23] or from a single high-resolution fractionation experiment [26]. A recent study impressively demonstrated interactome mapping across different mouse tissues [27]. In comparison to XL-MS, CoFrac-MS approaches that utilize high-resolution fractionation strategies, such as SEC, can obtain a much deeper proteome and interactome coverage, reaching > 1000 complexes composed of > 10,000 PPIs from a single two-condition experiment [28]. While coverage is still technically limited by the proteomic depth and dynamic range that can be captured by the selected MS technology, CoFrac-MS could, in principle, be used to simultaneously assess interactions occurring across the entire proteome.

Similar to CoFrac-MS, thermal proteome profiling (TPP) also offers a global strategy to assess PPIs and protein complexes. TPP is a method to systematically measure protein thermal stability and abundance [29]. Here, a proteomic sample is heated to increasing temperatures, thereby inducing protein denaturation. At each temperature, soluble proteins are quantified by high-resolution MS, a process which yields denaturation curves for all detectable proteins. TPP studies have shown that proteins that are part of the same protein complex show similar thermostability profiles [30,31,32]. While CoFrac-MS is expected to provide more depth and sensitivity in detecting changes in the composition of protein complexes and their abundance across conditions, TPP has the potential to identify more subtle changes in protein stability [33]. These could for example arise from allosteric regulation by small molecule binding to a protein complex, thus influencing its thermostability.

A further, more indirect strategy for investigating protein complexes in an untargeted and holistic fashion is to perform full proteome co-variation analysis. Here, data is typically not specifically acquired for protein complex analysis. The basis for full proteome co-variation studies are large MS datasets, which can be computationally leveraged to investigate co-variation of proteins. This strategy has been applied to identify evolutionarily conserved or highly variable protein complexes across species and individuals [34,35]. The sensitivity of such analyses largely depends on the number of samples at hand. Furthermore, it is important to keep in mind that co-variation (i.e., co-expression) is not necessarily equivalent to co-complex membership, but could also refer to functional, indirect interactions (for example, a transcription factor regulating the abundance of another protein). A recent study highlighted the potentially misleading observations from co-expression studies [27]. Nevertheless, full proteome co-variation analyses still provide an initial estimate of protein complex information, especially when combined with prior information and when the investigation of global patterns is the main interest.

In summary, MS-based workflows for interactome profiling can be categorised into targeted (AP-MS and proximity labelling) and untargeted (XL-MS, CoFrac-MS, TPP and full proteome co-variation MS) interactome acquisition strategies. The targeted approaches require a large number of reciprocal experiments to obtain a holistic view of the interactome of a single sample. In contrast, XL-MS, CoFrac-MS and TPP, in principle, enable the inference of a PPI network from a single sample in one experiment (typically including multiple MS runs). Full proteome co-variation analysis, by definition, cannot be performed on a single sample, but requires large MS cohorts of hundreds of individual samples. Here, multiplexed data acquisition strategies such as stable isotope labeling with amino acids in cell culture (SILAC) provide the only possibility to reduce the number of required MS runs. Differently to targeted interactome acquisition strategies, the proteome and interactome coverage of untargeted interactome screens is solely determined by current technical state of the art, but not by the acquisition concept itself.

## 3. Discovery and Hypothesis Driven Data Analysis Strategies

A major challenge in untargeted interactome screens, especially in CoFrac-MS, TPP and full proteome co-variation analysis, is to differentiate between signals originating from protein complexes and signals derived purely from random co-elution/co-variation. In this section I describe the two main approaches for analyzing data from untargeted interactome screens: (1) discovery- and (2) hypothesis-driven data analysis. Figure 2 illustrates their conceptual differences and summarizes their benefits and limitations.

The most widely applied data analysis strategy for untargeted interactome screens is the discovery approach. Similar to discovery proteomics, where the goal is the identification of a protein, discovery analyses in interactomics are focused on the identification of novel protein complexes in a given dataset. The analysis concept is based on calculating pairwise scores (for example, based on correlation) between all detected proteins, followed by a classification task that determines which protein-pairs likely represent a true interaction. The classification is commonly achieved by machine learning (ML) algorithms that are trained based on a positive and negative reference set of interactions [28,36,37]. Proteins in the resulting interaction network can be grouped into defined protein complexes by applying a graph partitioning algorithm. The inferred protein complexes can then be classified as either known or novel by cross-referencing to available complex databases. Previous studies have successfully implemented different versions of the discovery concept for analyzing interactome data, mainly derived from CoFrac-MS studies [23,24,25,38,39,40,41]. The key benefit of the discovery approach is that only limited prior knowledge is necessary (only enough to train the ML algorithm) and that novel protein complexes can be readily identified. In practice, high proteomic coverage coupled to limited biochemical resolution (determined by the resolution of fractionation strategies in CoFrac-MS, sampled temperatures in TPP and the number of individual samples in full proteome co-variation analysis) results in a high degree of random correlation of non-interacting proteins [42]. This has negative effects on the sensitivity and selectivity of purely discovery-based approaches. This challenge can be addressed by integrating information from multiple orthogonal fractionations [23] or datasets [43], but this comes at the cost of large experimental efforts.

Targeted (i.e., hypothesis-driven) data analysis strategies provide an alternative approach to address the challenges associated with untargeted interactome screens. Prior knowledge about known PPIs or protein complexes is leveraged to increase the sensitivity, selectivity, and consistency of quantitative information. In *complex-centric* data analysis, a protein complex database is used as prior information to directly test for evidence of specific protein complexes in the dataset at hand. Here, only a small fraction of the pairwise comparisons are necessary, thereby markedly reducing the search space and the risk of false positives. Complex-centric analysis of interactome data can be regarded as analogous to peptide-centric data analysis of proteomics data generated by data-independent acquisition (DIA) [44,45]. In peptide-centric DIA analysis, a spectral library is utilized to perform a targeted extraction of fragment-ion chromatograms from highly convoluted MS2 spectra. This improves the sensitivity, selectivity and consistency of peptide detections. In complex-centric interactome analysis, the protein complex database serves an equivalent purpose as the spectral library for DIA. Despite this general analogy, it is important to keep in mind that peptide to protein associations are much stronger than protein to complex associations. While spectral libraries in DIA commonly contain only high confidence identifications and are specific to a given experiment, PPI and protein complex databases are usually generic and information of various confidence could be included. The higher the confidence of protein complexes in the reference database and the more representative it is for a given experiment, the closer complex-centric analysis resembles the characteristics of the peptide-centric workflow. As stated above, previous studies have shown significant benefits in the sensitivity and selectivity of complex-centric approaches compared to discovery analyses [26]. However, this comes at the cost of not being able to identify completely novel protein complexes and being dependent on trustworthy prior knowledge, which might not always be available. Complex-centric analysis will therefore substantially benefit from databases with increasing confidence and modularity.

We recently applied complex-centric analysis to map the interactome of the human HEK293 cell line in a single CoFrac-MS experiment [26,42]. We used protein complexes annotated in CORUM [46], or derived via graph partitioning from BioPlex [9] and STRING [47], and searched for evidence of their presence in the CoFrac-MS data. False discovery rates were determined based on a set of randomly generated complex decoys. Complex-centric analysis has also been applied to analyze data generated by TPP [31]. Here, a set of 279 manually curated, largely non-overlapping protein complexes [48] was taken as prior information for a targeted analysis of subunit correlation in comparison to a random set of complex decoys. The same hypothesis set was applied in a study by Romanov et al., who analyzed protein complex co-variation across 11 full proteome datasets from humans and mice [34]. Stalder et al. further applied complex-centric analysis based on ortholog mapping to provide evidence that complex covariance profiles are conserved across species [35].

An alternative hypothesis-driven approach for untargeted interactome screens follows a *network-centric* rationale. Here, an entire PPI network is leveraged as prior information. Instead of looking at groups of proteins together (as is the case in complex-centric analysis), the network-centric approach evaluates individual edges in a network to test if the edge is supported by the given data, for example, by a positive correlation of intensities across fractions. The reference PPI network is thereby updated during the process, finally representing only condition-specific interactions. The main benefit of the network-centric analysis concept compared to discovery approaches is that its search space is significantly reduced, thus boosting sensitivity and selectivity. However, in contrast to complex-centric analysis, no protein complex inference (i.e., grouping of proteins to defined protein complexes) is conducted, which improves robustness against technical effects and allows prior knowledge to be less strictly defined, also including low confidence interactions in a PPI network. These properties make the network-centric approach particularly scalable across multiple conditions and larger datasets, where protein complex subunits may not be equally measurable across all conditions. We recently developed and applied a network-centric strategy for analyzing CoFrac-MS data across different cell cycle stages in the HeLa cell line [49]. Although prior knowledge is still required for the successful application of network-centric analysis, the approach also works well for predicted PPI networks such as PrePPI [50]. This indicates that network-centric analyses can be utilized to confirm predicted interactions using actual data. The network-centric concept is still new in proteomics-based interactomics, but these characteristics promise a high potential for it to gain increasing momentum in the future.

In summary, analysis strategies for untargeted interactome screens can be divided into discovery- and hypothesis-driven approaches. The main benefit of discovery approaches is that they enable the identification of novel complexes without requiring prior knowledge. However, they suffer from reduced sensitivity and selectivity in single experiments. Hypothesis-driven strategies overcome this challenge by using prior knowledge to reduce the search space and allow consistent detection and quantification of interactions and protein complexes across conditions. The main limitation of hypothesis-driven strategies is the necessity to have access to prior knowledge of sufficient depth and quality. Complex-centric analysis might require the use of graph partitioning algorithms, such as ClusterOne [51], to derive defined protein complex modules from a PPI network such as STRING [52]. Network-centric approaches can directly utilize such PPI networks and are thereby less dependent on variable parameter selection. Although both complex- and network-centric approaches benefit from high-confidence priors, strong protein co-variation patterns, especially in CoFrac-MS, can serve as important evidence to also verify low-confidence priors, for example derived from prediction tools. In this context it is important to note that experimental data acquisition and data analysis are not fully independent. The lower the resolution of the experimental data, the more confident the prior information has to be for deriving meaningful results. However, it is critical to keep in mind that poorly designed experiments at low resolution cannot purely be compensated by elaborate targeted analysis strategies. Importantly, discovery and targeted approaches are not mutually exclusive. Similar to recent attempts [28], large datasets can first be analyzed by a discovery approach to determine a set of tentative protein complexes, followed by targeted re-extraction and quantification of protein complexes. This is analogous to current hybrid approaches for DIA analysis, exemplified by directDIA in Spectronaut [53] or workflows that couple library generation by DIA-Umpire [54] with targeted extraction by OpenSWATH [55]. Such hybrid approaches have the potential to combine the benefits of both discovery and targeted analysis concepts and provide a promising future direction.

## 4. Protein Complex and PPI Databases

As discussed above, hypothesis-driven analysis approaches offer many benefits. However, their successful application depends on the availability of prior knowledge based on PPI or protein complex databases. Here, I review the most commonly applied databases, including the type of information they contain (for example, curated or predicted), the organisms they cover and their comprehensiveness. Please note that I cover only a selection of the most comprehensive, widely used and recently updated PPI databases (for an inclusive review see [56]). Table 1 summarizes information about the selected protein complex and PPI databases.

Protein complex databases generally include modules of physically interacting proteins that have been experimentally observed. The most commonly used, gold-standard database for protein complex information is the CORUM database, which contains 4274 manually-curated protein complexes across different mammals, mainly human (67%), mouse (15%) and rat (10%) [57]. An alternative protein complex database is the Complex Portal [58]. This covers fewer mammalian complexes compared to CORUM, but contains complex information for other organisms, for example, *A. thaliana*, *S. cerevisiae*, *E. coli*, *C. elegans* and *SARS-CoV-2*. HuMap is a third protein complex database, containing only human complexes. In contrast to CORUM and Complex Portal, HuMap is not manually curated, but contains protein complex information derived from the integrative analysis of over 15,000 mass spectrometry experiments covering AP-MS, proximity labelling and CoFrac-MS measurements [59]. HuMap covers 6969 protein complexes consisting of 57,178 unique interactions among 9968 human proteins.

Among available PPI databases solely based on experimental data, IntAct [60] and BioGrid [61] offer very comprehensive networks. IntAct contains 1,130,596 interactions among 119,281 proteins, mainly from human (61%), yeast (12%) and mouse (8%), but also covering other organisms across all domains of life [60]. IntAct provides all PPI contributions to the Complex Portal database discussed above [58]. BioGrid is another manually-curated PPI database, containing information about physical, as well as genetic, interactions between proteins [61]. BioGrid covers more than 70 species, with a total of 907,858 physical and 694,730 genetic, non-redundant interactions. In contrast to these organism-wide and multi-experiment databases, BioPlex is a resource of PPIs generated from a single experimental AP-MS study of human cell lines [10]. The most recent release, 3.0, contains 118,162 interactions among 14,586 proteins. At the other end of the spectrum, PrePPI is a large prediction-based PPI database, containing 1.35 million predicted interactions for ~85% of the human proteome [50]. Finally, STRING is the largest and most common PPI database, and integrates experimental and predicted PPI information for 5,090 different organisms across all domains of life, covering a total of 24.6 million proteins [52]. A key asset of STRING is its intuitive usability and the possibility to filter PPIs based on various confidence criteria, which will determine the overall confidence of the resulting network.

## 5. Considerations for the Selection of Interactome Acquisition and Analysis Approaches

Many different interactome acquisition and analysis approaches, as well as available prior knowledge, have been discussed, but the question remains how to decide between these different strategies. To guide the reader in experimental design, I here present different scenarios based on alternative research questions and provide advice for the most appropriate interactome acquisition and analysis approaches.

Targeted interactome acquisition approaches are usually the method of choice for studies aiming to investigate the interaction partners of a single protein, a specific pathway or a submodule in the cell. If stable, physical interactions are of interest, AP-MS should be used. If transient interactions or short, time-resolved processes are of interest, proximity labelling may be preferred. Given the constraints of keeping experimental costs and time in a feasible range, a trade-off must be made between the number of investigated bait proteins and the number of samples (for example, conditions).

XL-MS is the leading MS-based method for structural studies in which the interest is not only the protein complexes themselves, but also their three-dimensional structures and binding interfaces. However, system-wide XL-MS studies are still technically limited in their achievable proteome and interactome coverage [19]. Therefore, CoFrac-MS is currently the method of choice for proteome-wide screening of interactome rewiring across samples or conditions. In this context, different fractionation strategies should be selected (reviewed in [62]) depending on the desired resolution and type of protein complexes under investigation (for example, soluble or membrane bound). TPP is another alternative to CoFrac-MS. It could be of specific interest if the research question is to evaluate subtle changes in protein complex stability, for example upon small molecule perturbation. For MS data acquisition, DIA has proven superior to data-dependent acquisition (DDA) when label-free quantification is used [26]. Once data is acquired, the choice of an appropriate data analysis approach is crucial. If the focus of the research question is on the identification of novel complexes, or if prior information is sparce, a discovery approach should be selected. On the other hand, hypothesis-driven approaches are the preferred choice when confident detection and quantification of defined PPIs or protein complexes within a sample or across conditions is desired. The prerequisite of a hypothesis-driven analysis is the availability of appropriate prior knowledge. Table 1 summarized some of the most comprehensive and commonly used PPI and protein complex databases. To extend the scope of hypothesis-driven approaches, these existing databases can also be supplemented by appending manually-curated protein complex entries. Importantly, different experimental interactome acquisition strategies can also be combined to increase confidence in discovered PPIs and to reduce false positives. This can be achieved by integrating respective analysis results or by directly coupling workflows, exemplified by cross-linking coupled to CoFrac-MS [63].

The above-mentioned scenarios are geared towards studies that aim to identify or quantitatively compare interactomes across specific samples, with the capacity to acquire exclusive data for the analysis. For more general systems biology questions, where patterns rather than explicit examples are of interest (for example, concerning general complex co-variation across individuals or species), full proteome co-variation analysis can provide sufficient resolution to gain valuable insights. Importantly, systematic co-variation studies across full-proteome datasets can enable the inference of protein complex level information for cohorts with sample amounts that are too limited for specific interactome data acquisition, for example, in clinical studies. While discovery analysis in such datasets is hampered by an exploding search space, complex-centric analysis approaches have been shown to provide valuable results [34,35].

## 6. Summary and Future Perspectives

The identification of protein complexes and investigation of their dynamic rewiring across biological conditions is an active area of research that is of interest in both basic and translational science. To date, AP-MS remains the gold standard for generating high confidence interactome maps. However, the necessity to perform a separate experiment for every bait provides limited scalability for systems biology studies. Due to its multiplexed experimental design, CoFrac-MS has become a very promising strategy to perform comparative interactome screens. Technical improvements, especially to MS instrumentation, data acquisition and analysis, will continue to improve the proteome coverage achievable in single MS runs, exemplified by recent break-throughs with novel DIA technologies [64,65]. Recent developments in the LC setup for MS data acquisition, exemplified by micro-flow applications [66] or the Evosep One system [67], will further increase the throughput of interactome screens. When coupled with highly sensitive MS instruments that can operate on small sample amounts, CoFrac-MS studies could also collect fractions at higher frequency, thus increasing the resolution for protein complex analysis. Such technical improvements have a direct impact on the interactome coverage and throughput that can be achieved by CoFrac-MS and should promote its wider use in the future. The main data analysis strategy for CoFrac-MS data follows the discovery rationale. As highlighted above, this has the benefit of enabling the identification of novel complexes, but often comes at the cost of compromised sensitivity and selectivity. Alternative hypothesis-driven approaches that leverage prior information from public PPI and protein complex databases were developed and benchmarked only recently [26,49]. While the use of prior knowledge improves the sensitivity and selectivity of hypothesis-driven analysis, its applications are limited by the availability of appropriate databases. Over the last years, the number and comprehensiveness of PPI and protein complex databases grew markedly, including information generated by low-throughput experiments, high-throughput screens, and fully predicted PPI networks. It is expected that available interactome information will continue to grow and improve over time, thus providing increasingly confident and modular information that can be leveraged for hypothesis-driven analysis. Together with continuously improving experimental data, and increasing sample sizes of proteomic and interactomic studies, I predict that these developments will cause a paradigm shift towards hypothesis-driven data analysis in interactome screens. It will remain interesting to observe the directions in which the field will develop.

## Figures and Tables

**Figure 1 ijms-22-04450-f001:**
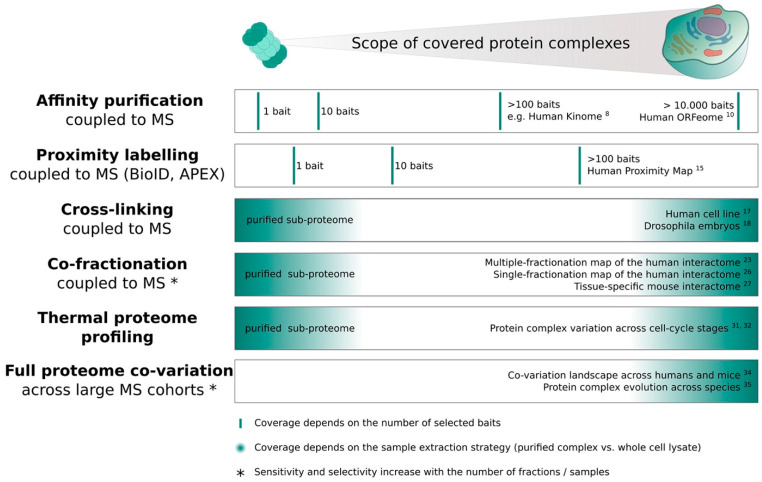
The scope of different mass spectrometry (MS)-based interactome screens. State of art MS-based approaches for investigating the interactome are summarized. Affinity purification and proximity labelling coupled to MS are targeted strategies for which a high coverage of protein complexes can only be achieved by performing hundreds to ten-thousands of single-bait experiments. The theoretically achievable coverage of cross-linking, co-fractionation and thermal proteome profiling strategies in turn depend on the purification level of the sample, e.g., purified sub-proteome (single complex or organelle) vs. whole cell extract. Finally, full proteome co-variation studies across large MS cohorts are usually not acquired specifically with the purpose of investigating protein interactions, but they can be leveraged to find global patterns of co-variation e.g., to investigate evolutionary conservation of complexes. Exemplary studies for the different approaches are indicated at their respective proteome coverage.

**Figure 2 ijms-22-04450-f002:**
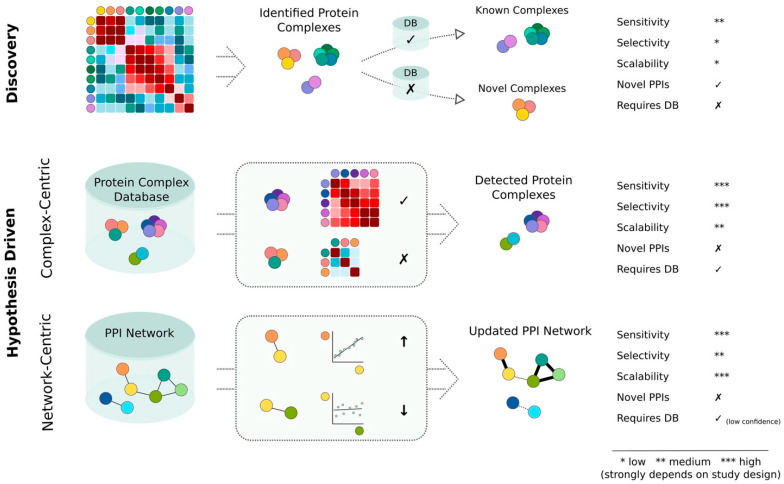
Discovery- vs. hypothesis-driven analysis concepts for interactome analysis. The two main strategies for analyzing untargeted interactome screens are discovery and hypothesis driven approaches. To focus on the conceptual differences, all approaches are illustrated by a correlation-based analysis. The colored circles indicate different proteins. Correlation matrices are illustrated by square grids with high correlations indicated in shades of red and low correlations indicated in shades of blue. In the context of this figure the term scalability refers to the ability of the method to provide consistent and comparable results at modular resolution across multiple datasets and conditions. Note that this figure only represents a schematic illustration of the different analysis concepts which can be implemented in various different flavors and which will also influence the compared performance metrics. The check signs indicate ‘yes’ and cross signs indicate ‘no’. The arrows indicate an increase (up) and decrease (down) in correlation.

**Table 1 ijms-22-04450-t001:** Summary of protein complex and PPI databases.

	DB	Information	Interaction type	Organisms	Size	Website
**Complexes**	CORUM 3.0	Manually curated from experimental data	Direct (physical) interactions	Human (67%) Mouse (15%) Rat (10%) & other mammals	4274 complexes based on 4473 genes (including 22% of human protein coding genes)	http://mips.helmholtz-muenchen.de/corum/ (accessed on 9 March 2021)
	Complex Portal (accessed 9 March 2021)	Manually curated from experimental data	Direct (physical) interactions	26 organisms across all domains of life	Human: 985 complexes Mouse: 740 complexes Yeast: 607 complexes *E. coli*: 321 complexes Others	http://www.ebi.ac.uk/complexportal (accessed on 9 March 2021)
	huMap 2.0	Integration of over 15,000 mass spectrometry experiments	Direct (physical) interactions (and proximity interactions)	Human	6969 complexes consisting of 57,178 unique interactions among 9,968 proteins	http://humap2.proteincomplexes.org/ (accessed on 9 March 2021)
**PPIs**	IntAct (accessed 11 March 2021)	Manually curated from experimental data	Direct (physical) interactions	Human (61%) Yeast (12%) Mouse (8%) & other organisms across all domains of life	1,130,596 interactions among 119,281 proteins	http://www.ebi.ac.uk/intact (accessed on 9 March 2021)
	BioGRID 4.3	Manually curated from experimental data	Direct (physical) interactions and genetic interactions	70 species	Budding yeast: 755,000 PPIs Fission yeast: 79,000 PPIs Human: 670,000 PPIs Worm: 29,000 PPIs Fly: 78,000 PPIs All other organisms: 300,000 PPIs	https://thebiogrid.org/ (accessed on 9 March 2021)
	BioPlex 3.0	Experimental	Direct (physical) interactions	Human	118,162 interactions among 14,586 proteins	https://bioplex.hms.harvard.edu/ (accessed on 9 March 2021)
	PrePPI	Predicted	Direct (physical) and indirect (functional) interactions	Human	PrePPI contains 1.35 million PPIs for ~85% of the human proteome	http://bhapp.c2b2.columbia.edu/PrePPI (accessed on 9 March 2021)
	STRING v11	Experimental & predicted	Direct (physical) and indirect (functional) interactions	5090 different organisms	>2000 million unique interactions among 24.6 million proteins	https://string-db.org/ (accessed on 9 March 2021)

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
