# Peer review of "Discovery–Versus Hypothesis–Driven Detection of Protein–Protein Interactions and Complexes"

_ijms, 2021, doi:10.3390/ijms22094450_

Round 1

Reviewer 1 Report

This is a really useful and comprehensive review.

L57: Remove one instance of ‘directly’.

L163: Full proteome co-variation analysis could be achieved in microorganisms, yeast or cell lines through the use of SILAC allowing limited amount of multiplexing. The question would be whether it would be a practical alternative.

One of the limitations is the chromatographic cycle time required to analyse each fraction from SEC by LC-MS/MS. If LC-MS/MS cycle time could be reduced to a few minutes, which has been demonstrated (Bian, Y., et al. (2020). "Robust, reproducible and quantitative analysis of thousands of proteomes by micro-flow LC-MS/MS." Nat Commun 11(1): 157 and http://dx.doi.org/10.1101/656793), then more discreet fractions (<1min) could be collected resulting in more points across the SEC peak for a given protein and more resolution for the complex. Microbore and microflow SEC would improve this resolution.

L208: multiplexed is often used to describe the mixing of multiple samples into a single experimental analysis. While I acknowledge that multiplexed MS2 spectra refers to fragments from multiple peptides in the same MS2 scan, a better term should be used.

L210-218: The problem described can be somewhat overcome through the idea of guilt by association provided by the SEC separation.

Paragraph L169 and then starting L262: The analysis strategies are discussed as being independent of the wet experiment but that is misleading. The aforementioned guilt by association provided by the SEC separation can differentiate whether a signal is from a complex. But if an unexpected protein is in the complex, orthogonal experiments need to be performed  to decided whether its presence is through non-specific, referred to as random on L171. Databases are only as good as the information in them, a point made earlier in the manuscript. Complex data analysis should not make up for poor experimental design.

Section 5: It should be noted that performing multiple experiments, such as CoFrac-MS and XL-MS, on the same sample provides a level of validation because the same interactions should be observed in both experiments, increased confidence.

Author Response

Comments and Suggestions for Authors

This is a really useful and comprehensive review.

L57: Remove one instance of ‘directly’.

I have removed one ‘directly’.

L163: Full proteome co-variation analysis could be achieved in microorganisms, yeast or cell lines through the use of SILAC allowing limited amount of multiplexing. The question would be whether it would be a practical alternative.

I have included a statement that multiplexing offers the possibility to reduce the number of samples required for full proteome co-variation studies.

Lines 186-187: “Here, multiplexed data acquisition strategies such as SILAC provide the only possibility to reduce the number of required MS runs.”

One of the limitations is the chromatographic cycle time required to analyse each fraction from SEC by LC-MS/MS. If LC-MS/MS cycle time could be reduced to a few minutes, which has been demonstrated (Bian, Y., et al. (2020). "Robust, reproducible and quantitative analysis of thousands of proteomes by micro-flow LC-MS/MS." Nat Commun 11(1): 157 and http://dx.doi.org/10.1101/656793), then more discreet fractions (<1min) could be collected resulting in more points across the SEC peak for a given protein and more resolution for the complex. Microbore and microflow SEC would improve this resolution.

This is an important point. I have included a short section discussing recent developments in LC-MS/MS throughput and their impact on CoFrac-MS experiments.

Lines 431-435: “Recent developments in the LC setup for MS data acquisition, exemplified by micro-flow applications [66] or the Evosep One system [67], will further increase the throughput of interactome screens. When coupled with highly sensitive MS instruments that can operate on small sample amounts, CoFrac-MS studies could also collect fractions at higher frequency, thus increasing the resolution for protein complex analysis.”

L208: multiplexed is often used to describe the mixing of multiple samples into a single experimental analysis. While I acknowledge that multiplexed MS2 spectra refers to fragments from multiple peptides in the same MS2 scan, a better term should be used.

I understand the reviewers concern and changed the term ‘multiplexed’ to ‘convoluted’.

L210-218: The problem described can be somewhat overcome through the idea of guilt by association provided by the SEC separation.

I addressed this comment together with the next point, by stating that strong co-variation signals in high-resolution data can be used to verify low-confidence priors. Please also see my answer to the next point.

Paragraph L169 and then starting L262: The analysis strategies are discussed as being independent of the wet experiment but that is misleading. The aforementioned guilt by association provided by the SEC separation can differentiate whether a signal is from a complex. But if an unexpected protein is in the complex, orthogonal experiments need to be performed  to decided whether its presence is through non-specific, referred to as random on L171. Databases are only as good as the information in them, a point made earlier in the manuscript. Complex data analysis should not make up for poor experimental design.

I agree with the reviewer’s concern that data acquisition and analysis are not independent and that complex data analysis should certainly not make up for poor experimental design. I addressed this comment in the manuscript in lines 315-323:

“Although both complex- and network-centric approaches benefit from high-confidence priors, strong protein co-variation patterns, especially in CoFrac-MS, can serve as important evidence to also verify low-confidence priors, for example derived from prediction tools. In this context it is important to note that experimental data acquisition and data analysis are not fully independent. The lower the resolution of the experimental data, the more confident the prior information has to be for deriving meaningful results. However, it is critical to keep in mind that poorly designed experiments at low resolution cannot purely be compensated by elaborate targeted analysis strategies.”

Section 5: It should be noted that performing multiple experiments, such as CoFrac-MS and XL-MS, on the same sample provides a level of validation because the same interactions should be observed in both experiments, increased confidence.

This is an important point and I added a statement to the manuscript addressing the reviewer’s comment.

Lines 405-409: “Importantly, different experimental interactome acquisition strategies can also be combined to increase confidence in discovered PPIs and to reduce false positives. This can be achieved by integrating respective analysis results or by directly coupling workflows, exemplified by cross-linking coupled to CoFrac-MS [63].”

Reviewer 2 Report

The author has represented an interesting and even intriguing review of methods for evaluation of protein-protein interactions (PPI) proteome-wide. Being experienced in this field, she provided a valuable view on the evolution of these methods from inside. I highly appreciate this work. At the same time, it may be improved by addressing some specific notes.

Major notes

  1. Before going into deepness of MS-based interactomics, it would be good to explain the author’s understanding of PPI and, especially, of the term ‘protein complex’.

How can we determine the protein complex formally? Is there a specific limit for Kd of binding between its components to define them as component of one complex and another? Mostly, the thermodynamic and kinetic parameters of interactions between components in such a ‘complex’ are undiscovered. E.g. a classical protein complex, such as mTORC1 is not well characterized from the point of view of binding parameters, but is defined as a functional complex by mutants and knockdowns. Please express your vision on this delicate matter.

  1. In the light of previous note, a complex-based approach deserves some criticism. I want to provide an example. In 2004, Prof. Yates III et al published a paper with one of the first applications of LC-MS/MS to get yeast proteome. They announced about 2,000 proteins. If you take, e.g., Andromeda and search these data with 1% FDR, you get 200 proteins. The same is with databases of PPI, which were produced ten years ago. How can we use these data to formulate “complexes” we later check in new data? I often use STRING and need manually curate all connections which are formed in its “hairballs”. How to do this in holistic analysis? You suggested a smart analogy of a complex-based approach with DIA with spectral library. However, the spectral library data are more reliable than DIA itself in terms of peptide-spectrum matching. In contrast, PPI database normally is less reliable than new data got, for example, with CoFrac-MS. Please discuss this issue in more detail.

  1. Thermal profiling for PPI. Another perspective approach which was originally used in pharma to find protein partners for xenobiotics and recently used for holistic PPI screen. Please add use of variants of thermal protein profiling to your text, cf. doi 10.1126/science.aan0346 and others.

  1. PPI holistic works literally demonstrate a brute force approach with many experiments and a database in the end. Unfortunately, that is often it. Are there examples where a PPI work led to the successful application in biomedicine? If exist, they would be a good final story for the review.

Minor notes

  1. Check ref. 27, no journal there.
  2. Fig.2 has some empty squares across Novel PPIs and Requires DB. Please fill.

Author Response

The author has represented an interesting and even intriguing review of methods for evaluation of protein-protein interactions (PPI) proteome-wide. Being experienced in this field, she provided a valuable view on the evolution of these methods from inside. I highly appreciate this work. At the same time, it may be improved by addressing some specific notes.

Major notes

  1. Before going into deepness of MS-based interactomics, it would be good to explain the author’s understanding of PPI and, especially, of the term ‘protein complex’.

How can we determine the protein complex formally? Is there a specific limit for Kd of binding between its components to define them as component of one complex and another? Mostly, the thermodynamic and kinetic parameters of interactions between components in such a ‘complex’ are undiscovered. E.g. a classical protein complex, such as mTORC1 is not well characterized from the point of view of binding parameters, but is defined as a functional complex by mutants and knockdowns. Please express your vision on this delicate matter.

This is a very good point and I included a definition of what I consider a ‘PPI’ and ‘protein complex’ in the introduction.

Lines 31-37: “Importantly, this review mainly focuses on the physical interactome. Thus, a PPI generally refers to a physical interaction between two proteins, unless otherwise stated. A PPI network consequently represents a set of proteins that are connected to each other via PPIs. A protein complex, in turn, refers to a stable group of physically interacting proteins. However, the physicochemical boundaries for differentiating a stable protein complex from more dynamic functional modules in the cell still remain to be defined. In context of this review, clusters of co-regulated PPIs are termed protein complexes.”

  1. In the light of previous note, a complex-based approach deserves some criticism. I want to provide an example. In 2004, Prof. Yates III et al published a paper with one of the first applications of LC-MS/MS to get yeast proteome. They announced about 2,000 proteins. If you take, e.g., Andromeda and search these data with 1% FDR, you get 200 proteins. The same is with databases of PPI, which were produced ten years ago. How can we use these data to formulate “complexes” we later check in new data? I often use STRING and need manually curate all connections which are formed in its “hairballs”. How to do this in holistic analysis? You suggested a smart analogy of a complex-based approach with DIA with spectral library. However, the spectral library data are more reliable than DIA itself in terms of peptide-spectrum matching. In contrast, PPI database normally is less reliable than new data got, for example, with CoFrac-MS. Please discuss this issue in more detail.

 I agree with the reviewer’s concern and extended the discussion of this topic, see edits in lines 234-248 and 312-323.

“In complex-centric interactome analysis, the protein complex database serves an equivalent purpose as the spectral library for DIA. Despite this general analogy, it is important to keep in mind that peptide to protein associations are much stronger than protein to complex associations. While spectral libraries in DIA commonly contain only high confidence identifications and are specific to a given experiment, PPI and protein complex databases are usually generic and information of various confidence could be included. The higher the confidence of protein complexes in the reference database and the more representative it is for a given experiment, the closer complex-centric analysis resembles the characteristics of the peptide-centric workflow. As stated above, previous studies have shown significant benefits in the sensitivity and selectivity of complex-centric approaches compared to discovery analyses [26]. However, this comes at the cost of not being able to identify completely novel protein complexes and being dependent on trustworthy prior knowledge, which might not always be available. Complex-centric analysis will therefore substantially benefit from databases with increasing confidence and modularity.”

“Complex-centric analysis might require the use of graph partitioning algorithms, such as ClusterOne [51], to derive defined protein complex modules from a PPI network such as STRING [52]. Network-centric approaches can directly utilize such PPI networks and are thereby less dependent on variable parameter selection. Although both complex- and network-centric approaches benefit from high-confidence priors, strong protein co-variation patterns, especially in CoFrac-MS, can serve as important evidence to also verify low-confidence priors, for example derived from prediction tools. In this context it is important to note that experimental data acquisition and data analysis are not fully independent. High-resolution data from CoFrac-MS could for example be used to verify low-confidence protein complex hypotheses. The lower the resolution of the experimental data, the more confident the prior information has to be for deriving meaningful results. However, it is critical to keep in mind that poorly designed experiments at low resolution cannot purely be compensated by elaborate targeted analysis strategies.”

  1. Thermal profiling for PPI. Another perspective approach which was originally used in pharma to find protein partners for xenobiotics and recently used for holistic PPI screen. Please add use of variants of thermal protein profiling to your text, cf. doi 10.1126/science.aan0346 and others.

I thank the reviewer for pointing out that thermal proteome profiling (TPP) was missing from the review. I have added the TPP workflow description and examples to the text and Figure 1.

Lines 152-163: “Similar to CoFrac-MS, thermal proteome profiling (TPP) also offers a global strategy to assess PPIs and protein complexes. TPP is a method to systematically measure protein thermal stability and abundance [29]. Here, a proteomic sample is heated to increasing temperatures, thereby inducing protein denaturation. At each temperature, soluble proteins are quantified by high-resolution MS, a process which yields denaturation curves for all detectable proteins. TPP studies have shown that proteins that are part of the same protein complex show similar thermostability profiles [30–32]. While CoFrac-MS is expected to provide more depth and sensitivity in detecting changes in the composition of protein complexes and their abundance across conditions, TPP has the potential to identify more subtle changes in protein stability [33]. These could for example arise from allosteric regulation by small molecule binding to a protein complex, thus influencing its thermostability.”

TPP was also addressed in all relevant sections of the manuscript where the different interactome data acquisition approaches are discussed.

  1. PPI holistic works literally demonstrate a brute force approach with many experiments and a database in the end. Unfortunately, that is often it. Are there examples where a PPI work led to the successful application in biomedicine? If exist, they would be a good final story for the review.

I agree with the reviewer’s comment. However, I am not aware of any specific large-scale PPI work that directly led to a clinical application to date. While many studies exemplify the merit of their interactome screen with novel biological findings that are (sometimes) orthogonally validated, I cannot pinpoint a specific example where this led to a successful clinical application. However, the lack of direct biomedical impact is a common shortcoming of many ‘omics’ approaches. I generally view omics studies as screens to derive information about global characteristics of the biological system and to identify promising candidates that can be followed-up on. I expect that interactome studies will soon yield sufficiently promising results that will grant in depth clinical follow-ups. Once available at a sufficient scale, I expect interactome screens to also be used for the systematic screening of e.g. drug perturbations. I am overall optimistic that what nowadays often results in a database and systems biology interpretations will soon be leveraged to gain specific biomedically relevant information, similar to the trend in proteomics in general.

Minor notes

  1. Check ref. 27, no journal there.

Reference 27 is a paper preprint on bioRxiv. I have updated the reference.

  1. Fig.2 has some empty squares across Novel PPIs and Requires DB. Please fill.

I could not observe this effect in my version of the manuscript. It might have been a pdf conversion error. I converted the file again and hope the issue is solved now.

Round 2

Reviewer 2 Report

No more comments, thank you for this work.